

# Do pheromone traps help to reduce new attacks of *Ips typographus* at the local scale after a sanitary cut?

Alexandre Kuhn, Louis Hautier and Gilles San Martin

Crops and Forest Health Unit, Life Sciences Department, Walloon Agricultural Research Centre, Gembloux, Belgium

## ABSTRACT

The spruce bark beetle, *Ips typographus*, is causing severe economic losses during epidemic phases triggered by droughts and/or windstorms. Sanitation felling and salvage logging are usually the most recommended strategies to limit the damages. However, any additional control method to limit the economic impact of an outbreak would be welcome. In this respect, the efficiency of pheromone trapping is still controversial or poorly documented. In this 2-year study (2020–2021), at the peak of a severe outbreak in Belgium, we quantified the wood volume and presence/absence of new attacks at 126 sites attacked during the previous year and within 100 m from the initial attack. Each site was randomly allocated to one of three treatments: (1) three crosstraps baited with pheromones, (2) one tree-trap baited with pheromones and treated with an insecticide and (3) control sites with no trapping device. The attacked trees of the previous year were all cut and removed before the start of the experiment and newly attacked trees were removed as they were detected. The trapping devices were only active during spring to target overwintering bark beetles that might have escaped the sanitation cuts and to limit the risk of attracting dispersing beetles from outside the patch during the summer. We found a strong decrease of the attacks relative to the previous year in all treatments, including the controls (more than 50% of the control sites had no new attacks). There was no relationship between the new attacks and the attacks of the previous year. In both years, new attacks were more frequent (presence/absence) in sites with crosstraps (95% Confidence Interval [56–84%] of the sites with new attacks) than in sites with a tree-trap (26–57% - $p = 0.02$) and to a lesser extent than in control sites (32–63%, $p = 0.08$). In 2020, the attacked volumes were slightly higher in sites with crosstraps (95% Confidence Interval [3.4–14.2 m³]) than in control sites (0.2–3.5 m³, $p = 0.04$) and no significant difference was found with tree-trap sites (1.1–6.2 m³, $p = 0.38$). In 2021, there were no significant differences between the volumes attacked in the control sites (1.8–9.4 m³), crosstraps sites (0.9–6.4 m³) and tree-trap sites (0–2.5 m³). Overall, we found no evidence in favor of the efficacy of pheromone trapping during spring to reduce economic damages at the local scale when combined with sanitation felling and during a severe outbreak. The use of baited crosstraps could even be hazardous as it seemed to increase the occurrence of new attacks probably by attracting bark beetles but failing to neutralize them.

Corresponding author
Alexandre Kuhn,
a.kuhn@cra.wallonie.be

# INTRODUCTION

The spruce bark beetle, *Ips typographus* (L.), is a phloem-feeding insect native to Eurasia which typically feed and breed on weakened or wind-felled Norway spruces (*Picea abies* Karst.). Its populations periodically erupt into large-scale outbreaks after severe droughts and windstorms, switching from stressed trees to healthy ones. This beetle is considered as a keystone forest species and an ecosystem engineer as it profoundly alters forest structure by promoting heterogeneity in forest landscapes, which increases biodiversity (*Müller et al., 2008*; *Lehnert et al., 2013*; *Beudert et al., 2015*). However, it is also regarded as the most important forest pest in Europe due to its major impact on wood economy (*Grégoire & Evans, 2004*; *Wermelinger, 2004*).

Unprecedented outbreaks have marked European spruce forests in recent decades and climate change is expected to further increase the frequency and the magnitude of these events (*Marini et al., 2012*, *2017*; *Mezei et al., 2017*). The modification of the current cultural practices and the diversification of tree species within the stands are promising ways to mitigate bark beetles disturbances in the long run (*Jactel et al., 2009*; *Griess et al., 2012*; *Dobor, Hlásny & Zimová, 2020*; *Jactel, Moreira & Castagneyrol, 2021*). However, switching to more resilient forests takes time and *I. typographus* outbreaks should be properly managed in existing spruce stands to secure wood supply and mitigate socio-economic losses.

Currently, short-term strategies to limit *I. typographus* population growth mainly rely on two complementary methods: salvage logging and sanitary felling (*Wermelinger, 2004*; *Fettig & Hilszczański, 2015*). Salvage logging aims to remove highly attractive breeding material, such as wind-felled timbers, before it is colonized by bark beetles. Sanitary felling relies on the regular search for and removal of already infested trees. Although sanitation felling is considered the most effective direct control approach to reduce tree mortality (*Wichmann & Ravn, 2001*; *Wermelinger, 2004*; *Stadelmann et al., 2013*), it has some disadvantages. To be effective, infested trees have to be harvested before the emergence of the next generation, which usually takes between 4 and 6 weeks, depending on temperatures (*Wermelinger & Seifert, 1999*). This deadline is sometimes difficult to meet as not all infested trees show clear symptoms. Furthermore, in epidemic phases, interventions are often limited due to human resources shortage to deal with the overwhelming amount of timber to collect. In addition, detection is highly time consuming, logging is not always possible in remote places and it can remove natural enemy communities building in infested trees.

An alternative method of direct control is to trap and kill flying adults. Since the 19[th] century (*Pfeil, 1827*), foresters have taken advantage of the natural attractiveness of voluntarily felled trees and logs to concentrate and neutralize bark beetles. As for sanitation felling, natural tree-traps require regular attention to harvest or debark them before immature stages achieve their cycle and fly away, resulting in increased numbers of

beetles in the environment. Furthermore, the attractiveness of natural tree-traps may be masked in the presence of stressed standing trees and the beetles may spread to the latter. Finally, tree-traps can only capture a limited number of beetles before they become unattractive. In the late 70s, the aggregation pheromone blend of *I. typographus* was identified (*Bakke, Frøyen & Skattebøl, 1977*). This gave a new impetus in bark beetle management research and a diverse array of pheromone-baited traps were developed and tested. Mass-trapping was readily adopted as part of integrated pest management programs and thousands of baited traps were deployed in Europe to control *I. typographus* outbreaks. There is an extensive literature comparing the efficiency of different trap types and attractants in terms of the number of beetles caught (*Drumont et al., 1992*; *Raty et al., 1995*; *Galko et al., 2016*; *Holuša et al., 2017*; *Blaženec, Majdák & Jakuš, 2021*; *Heber et al., 2021*; *Lindmark, Wallin & Jonsson, 2022*). Catches appear to be quite variable and highly influenced by various factors such as weather conditions, sun exposure, trap density and disposition or bark beetle population levels (*Bakke, 1992*; *Lobinger & Skatulla, 1996*; *Galko et al., 2016*). Despite the impressive number of beetles caught by mass-trapping campaigns (*e.g.* 7.8 billion in Sweden between 1979 and 1980; *Bakke, 1989*), a growing number of scientists and practitioners have questioned the effect of trapping in the effective reduction of *I. typographus* population and damages. One claim against mass-trapping is the negligible share of the bark beetle population actually caught during an outbreak with a reasonable number of traps (*Weslien & Lindelöw, 1990*; *Weslien, 1992*; *Grégoire et al., 1997*; *Grégoire & Nageleisen, 2019*). With a high trapping intensity, it was estimated that between 5% (4 traps/100 ha) and 30% (15 traps/100 ha) of the population could be caught (*Weslien & Lindelöw, 1990*; *Weslien, 1992*). *Drumont et al. (1992)* found that, on average, each tree-trap (baited with pheromone and sprayed with insecticide) captured less than half the hibernating population from a single previously infested tree. Furthermore, most marked-recapture experiments indicated that only a small proportion (up to 10%) of released beetles (newly emerging or re-emerging, depending on the studies) respond to pheromones from local baited-traps (*Weslien & Lindelöw, 1990*; *Zolubas & Byers, 1995*; *Franklin, Debruyne & Gregoire, 2000*; *Dolezal, Okrouhlik & Davidkova, 2016*; but see *Duelli et al., 1997* for recaptures up to 35%). *Grégoire & Nageleisen (2019)* estimated that, in the absence of effective sanitary felling, 20 tree-traps or 60 artificial pheromone traps would be necessary to neutralize this 10% fraction of a 100 m³ infestation spot susceptible to perform local attacks. These results – although based on indirect measurements and several approximations that could lead to large uncertainties – suggest that traps alone are not able to contain bark beetle outbreaks at the landscape scale. Nevertheless, combined with other control measures (like sanitary felling), pheromone traps might contribute to reduce the economic damage caused by *Ips typographus* at a more local scale, for example by eliminating overwintering adults that could start new attacks in the spring close to previously infested spots (*Grégoire & Nageleisen, 2019*). During severe outbreaks, a simple reduction of 10–20% of the spruce volumes attacked might still be an important economic gain that should be balanced with the environmental and economic costs of trapping. However, this should be properly evaluated by direct measures of damage reduction rather than indirect measures (like number of insects caught). Yet, the scientific literature seems

to be rather poor in this respect, which is surprising given the economic importance of the issue.

Some studies do report the evolution of *I. typographus* damages over time in the presence of trapping devices. Unfortunately, most are case studies lacking control sites without traps or with a very limited number of controls (one or two) located in areas differing from the experimental areas in many aspects. The lack of a proper comparison point prevents any conclusions to be drawn from these studies on the real contribution of trapping to damage reduction (*Abgrall & Schvester, 1987*; *Bakke, 1989*; *Vité, 1989*; *Bombosch & Dedek, 1994*; *Jakuš, 1998*, *2001*; *Pfister & Hueber, 2008*). To the best of our knowledge, only a single published study properly compared damage with or without traps (*Faccoli & Stergulc, 2008*). They selected 24 previously attacked spots and monitored the volume of new attacks in the presence (pheromone slot-traps, standing trap-logs and horizontal trap-logs; six sites each) or in the absence of trapping devices (six control sites) during the next year. They found a dramatic decrease in new attacks (average reduction of about 80% relative to the previous year) in all sites with traps (whatever the trap type), while the level of new attacks remained high in the control sites. The results of this study suggest that trapping at a local scale may induce a strong economic gain at the stand level, even though its effect on the total bark beetle population is assumed to be negligible (see above).

In 2018, a massive *I. typographus* outbreak emerged in Belgium, triggered by repeated droughts and a winter windstorm (*OWSF, 2018*). Warm and dry conditions favored bark beetle development and the appearance of a third annual generation, leading to unprecedented population levels and dramatic economic losses. In this context, the controversy over trapping efficiency resurfaced among forest managers as any additional control method to limit the impact of the epidemic would be welcome. In this study, we evaluated the potential of pheromone trapping for damage reduction in previously infested patches during a severe outbreak in the Belgian landscape and climatic context. We regarded trapping as a complementary control method to sanitation felling to reduce local attacks and attenuate economic losses. Our objective was not to test the efficiency of large scale mass-trapping to curb the outbreak. During a 2-year experiment, we assessed the effect of two trapping modalities, panel crosstraps and tree-traps, relative to control sites without trapping devices. To maximize the chances of trapping success while minimizing the risks, traps were only left in place during the spring to catch the overwintering spruce bark beetles, when the population level at the landscape scale is the lowest and when the trees have replenished their water reserves. The rationale behind this choice was to improve the efficiency of sanitary cuts with the objective of extinguishing local infestation patches by capturing as much as possible of all remaining bark beetles from the first (overwintering) generation that could have escaped the sanitation felling from the previous year. At the same time, by removing the traps before the emergence of the second generation in summer, we wanted to limit the risk of attracting dispersing beetles from outside the patch at a time when population level at the landscape scale is at its highest (*Baier, Pennerstorfer & Schopf, 2007*) and the risk of saturating the traps is higher.

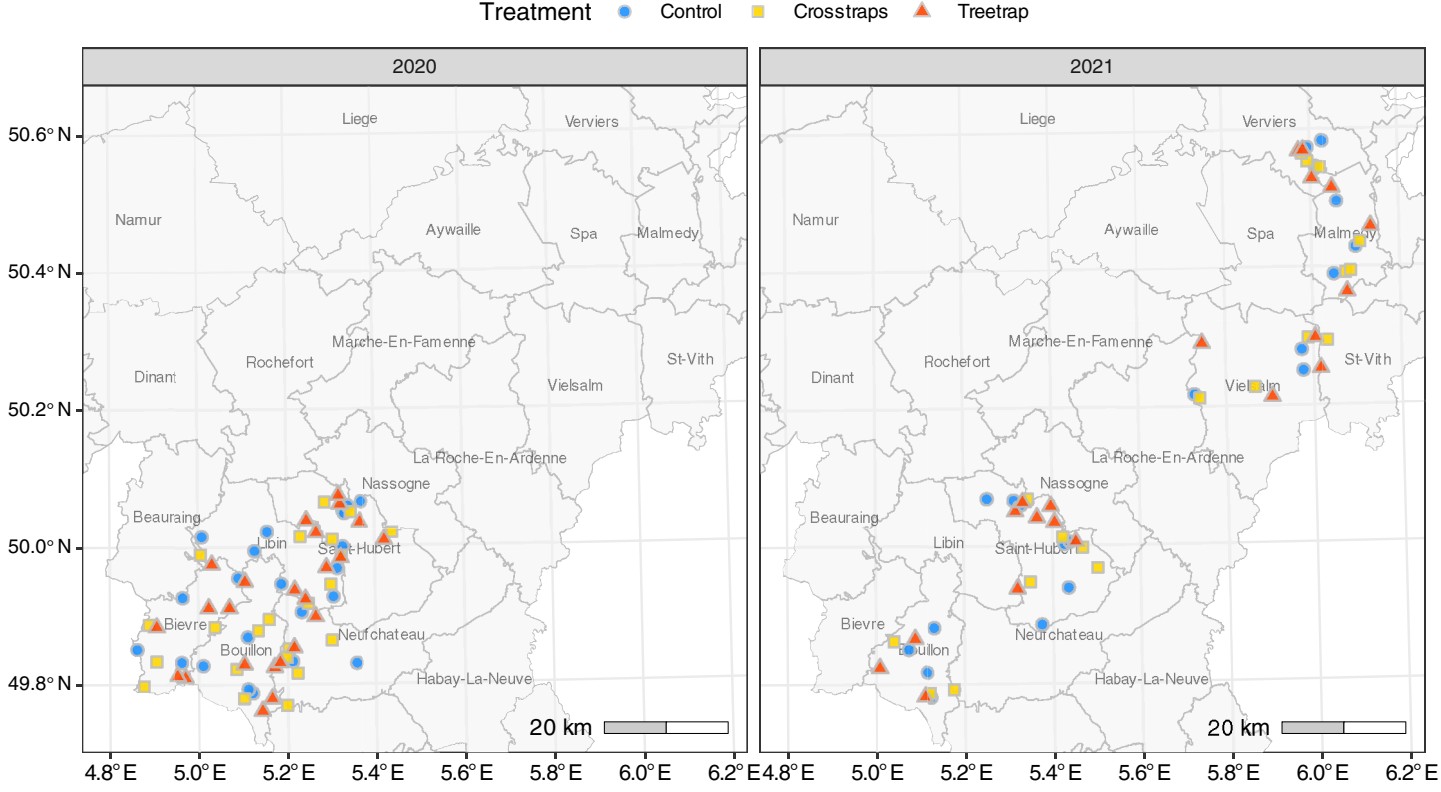

**Figure 1  Distribution of the sites for each treatment.** The administrative units drawn on the map correspond to public forest management districts.

## MATERIALS AND METHODS

### Experimental sites

Experimental sites consisted of infested patches within spruce stands with active bark beetle attacks during the summer and/or autumn of the year preceding the year of the experiment (attacks in 2019 for the 2020 experiment and attacks in 2020 for the 2021 experiment). We only selected sites with relatively circumscribed attacks and without other infested patches active during the previous year within a 250 m radius. In all selected sites, the forest officers had carried out a sanitary clear-cut to remove the trees attacked the previous year from the forest before the next spring.

A total of 126 sites were included in the experiment: 68 in 2020 and 58 in 2021. All sites were located in the Belgian Ardennes, a natural region where the ecological conditions are generally considered as suitable for Norway spruces by local foresters (*Petit et al., 2017*; Fig. 1). Ninety-three percent of the sites (117/126) were located in public forest stands, across seven forest districts of the Department of Nature and Forest. In 2021, nine sites were located in private forest stands managed by a single owner.

### Experimental set-up

Three treatments were tested: (1) artificial trapping device (hereafter referred to as "crosstraps"; Fig. 2A), (2) tree-trap device ("tree-trap"; Fig. 2B) and (3) the absence of any

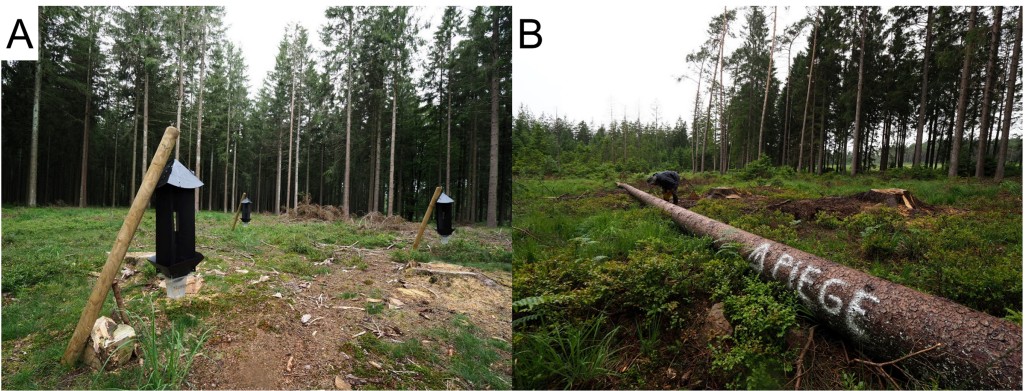

**Figure 2 Illustration of experimental devices.** (A) Crosstraps device. (B) Treetrap device.

trapping device ("control"). For tree-trap sites, a single healthy tree was felled, unbranched and placed in the approximate center of the clear-cut area of the patch, as far as possible from healthy trees of the clear-cut edges. Three Ipsowit® pheromone dispensers (Witasek, Austria) were placed evenly along the colder side (North or East if possible) of the tree-trap. The bark of the tree-trap was sprayed with an insecticide solution (1 g of lambda-cyhalothrin diluted in 1 L of water; Karate Zeon®, Syngenta, Belgium) under dry weather conditions and until the bark was soaked. For crosstrap sites, three cross-vane panel traps (Witasek, Austria) were hung on wooden stakes about 1.5 m high.

The crosstraps were arranged in line or in triangle (depending on the clear-cut shape), 3 to 10 m apart, and placed approximately in the center of the clear-cut, as far as possible from healthy trees of the clear-cut edges. Each crosstrap was supplied with a single Ipsowit® pheromone dispenser (Witasek, Austria). Trap collectors were filled with water.

For control sites, no trapping device was set-up. Whatever the size of the initial infested patch, the number of tree-traps was always one per site and the number of crosstraps was always three per site.

To limit site heterogeneity between treatments, the treatments were (as far as possible) evenly distributed among and randomly distributed within forest districts subunits (Fig. 1). These subunits, corresponding to smaller geographic entities, usually present similar environmental conditions, and are managed by a single forest officer.

The trapping devices (tree-traps and crosstraps) were set-up at the beginning of April, before the first *Ips typographus* flights, and removed at the beginning of July. We focused on the first – overwintering – generation to limit the risk of trap saturation with too many beetles from the second generation (up to three generations per year in Belgium during this outbreak). The pheromone dispensers were replaced after 6 weeks (mid-May) and the insecticide treatment of the tree-traps was renewed at the same period. The collectors of the crosstraps were emptied approximately every week by forest officers. Throughout the study period (2020 and 2021), forestry officers inspected the sites on a regular basis as they would normally do in any forest stand. New infestations were treated by sanitary felling (as
recommended for bark beetle management in Belgium) for all treatments (tree-trap, crosstraps and control).

## Number of insects captured by the crosstraps

To monitor the phenology of *I. typographus* flights and check that the traps were working, the number of bark beetles captured by the crosstraps was estimated based on their fresh volume. To estimate the average number of beetles per ml, we counted the number of individuals in 24 subsamples of trap captures with volumes ranging from 4 to 40 ml. The average number of beetles per ml showed little variation: 38 ± 4.4 individuals/ml (average ± SD). The weekly volume of bark beetles captured in each collector was measured. The samples with a volume ≥ 5 ml were converted with this volumetric method. For samples with a volume < 5 ml, the exact number of bark beetles was counted or roughly assumed to be equal to 100 bark beetles. In 2020, bark beetle volumes were only available for the Bièvre forest district (five crosstrap sites).

## Attack data acquisition

In 2020, new attacks on healthy trees were recorded by forest officers in three areas increasingly distant from the center of the clear-cut for each site: 0–50 m, 50–100 m or beyond 100 m from the center (but inside the same spruce stand). In 2021, new attacks were only recorded within a 100 m radius from the center of the initial clear-cut as most new attacks occurred close to the previously infested patch (see Results—Attacks at various distances from the patch center). This is also consistent with previous studies indicating that between 65 and 85% of new attacks occur within 100 m from old attacks (*Wichmann & Ravn, 2001*; *Kautz et al., 2011*; *Potterf et al., 2019*). When the clear-cut is very large and encompasses a large part of the 50 m radius zone, the number of trees remaining in this zone can be low and we therefore chose to merge the 0–50 m and 50–100 m zones. Although the traps were only active in spring (see above), we recorded the wood volume of newly attacked trees over the entire period of *I. typographus* activity (from spring to winter). The evolution of the attacks throughout the year was not recorded; only the final wood volume and the number of trees attacked were available.

## Data analyses

The statistical analyses and graphs were performed with the R programming language (*R Core Team, 2022*). All the data and R scripts needed to reproduce our results are available in a public figshare repository (see the Data Availability section).

### Sites homogeneity between trapping treatments

Before investigating the effect of trapping on new attacks, we checked that the sites attributed to the three treatments were similar on average in terms of environmental conditions, based on several variables describing the sites: number of trees and wood volume initially attacked in the patch, stand age, slope and orientation of the stand, clear-cut area of the patch, remaining spruce area around the patch, ecological suitability of the stand for Norway spruce and weather variables during the previous winter and during the period of activity of the bark beetles (temperature, precipitations, solar irradiance,

relative humidity and wind speed) (detailed description of the variables in the supplements, section 2.2). Basically, we compared the quantitative descriptors with simple ANOVA and the qualitative descriptors with chi-squared tests with simulated $p$-values (to avoid problems with categories with no or few observations).

### Impact of trapping treatments on new attacks

We used a linear model (ANCOVA) to compare the new volume attacked between treatments with log(x+1) transformed volume as response and year, treatment and their interaction as qualitative explanatory variables. We also added two continuous explanatory variables to the model: the initial volume attacked in the patch (during the previous year) and the area of spruce available for new attacks within a radius of 100 m around the center of the initial patch. This latter variable therefore represents the remaining quantity of resources available for new attacks within the monitored radius. Both additional covariates were square root transformed and centered on their mean value. They allow us to control statistically the variability between the initial patches for these two parameters that could influence the volume of new attacks. We then computed all pairwise comparisons between the three treatments with $p$-value correction (*Post-hoc* tests), single-step method from multcomp R package, (*Hothorn, Bretz & Westfall, 2008*; *Bretz et al., 2010*) and we report the 95% confidence intervals around the values predicted by the model. Because new volumes of 0 m³ were frequent, we also compared the proportion of sites with or without new attacks (presence/absence data) between the three treatments with a Generalized Linear Model (GLM) with binomial distribution. The rest of the analysis is similar to the linear model for volumes except the use of Likelihood Ratio (LR) tests for the main "analysis of deviance" table instead of the classical analysis of variance table.

Residual plots were used to check the conditions of application of the model (mostly: linearity, variance and distribution of the residuals, outliers) and guided the choice of the logarithmic and square root transformations (*Fox, 2002*; *Zuur, Ieno & Smith, 2007*; see Supplements, sections 3.1.5 and 3.2.5). We also checked for the absence of spatial correlation between the residuals of the model with spline correlograms (R package ncf, *Bjornstad, 2020*).

## RESULTS

### Homogeneity of the sites

We found no statistically significant differences between the three treatments in terms of site characteristics except for the site slope (see Supplements, section 2.2 for details). Tree-trap sites tended to have null slopes more frequently and low, medium or steep slopes less frequently than the other treatments (chi squared test with simulated $p$-value: $\chi^2 = 12.86$, $p = 0.045$). Therefore, the sites were globally very similar between the treatments.

### Number of insects captured in the crosstraps

In 2021, we captured between 8,000 and 45,000 individuals in each crosstrap site over an average period of 80 days (sum of the three crosstraps, between 2,000 and 22,000

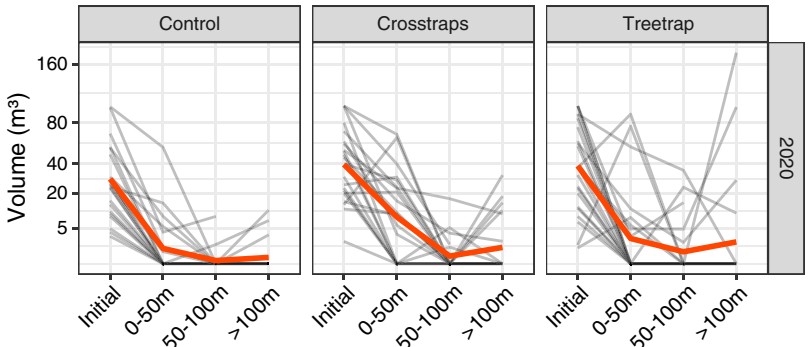

**Figure 3 Comparison of the wood volume of new attacks relative to their distance from the initial patch in 2020.** Each gray line represents the volume attacked in one site in 2020 at different distances from the initial patch (Initial = volume attacked during the previous year *i.e.*, 2019). The red line represents the average value. The new attacks tend to be concentrated close to the initial attack from the previous year. The attacks between 0–50 m and 50–100 m have been pooled in the final analyses (see Fig. 4).

individuals for each trap). In 2020, we obtained an estimate for only five sites from a single forest district, with 66,000 to 118,000 individuals captured per site (between 27,000 and 45,000 per trap).

We found no association between the number of individuals captured and the volume of new attacks on crosstraps sites in 2021 (log-log linear regression: slope = 0.29, s.e. = 0.48, t = 0.61, *p* = 0.55 – see Supplements, section 2.3.3).

## Attacks at various distances from the patch center

In 2020 only, we measured the volumes of new attacks at three distances from the center of the initial patch: 0–50 m, 50–100 m and beyond 100 m but within the spruce stand (Fig. 3).

The highest level of attacks was observed within 0–50 m from the patch center and decreased between 50 and 100 m. Beyond 100 m, the average volume attacked is lower than between 0–50 m but in a few sites we observed high levels of attacks that might be independent of the initial patch (Fig. 3).

We decided to sum the attacks between 0 and 100 m which are more likely to be related to the initial attack in the patch and also because the volumes of attacks between 0 and 100 m are highly correlated (on log transformed values) with both the volumes attacked between 0–50 m (r = 0.95) and the volumes of the whole stand (r = 0.92). In 2021, we measured only the levels of attacks between 0 and 100 m.

Surprisingly, the correlations between the new attacks and the initial volume attacked during the previous year are low (between r = 0.07 for volumes beyond 100 m and r = 0.24 for the attacks between 50 and 100 m; see Supplements, section 2.4.2 for details).

## Decrease of new attacks relative to the initial volume attacked

For both years of the study, we observed a strong decrease of the new volumes attacked relative to the initial attacks of the previous year in all treatments including the control sites without any traps (Fig. 4). In many sites, there were simply no new attacks (*e.g.*, 62%

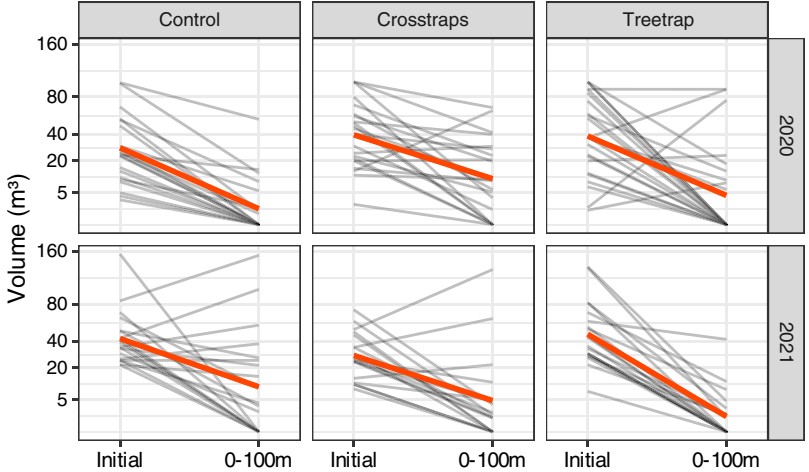

**Figure 4 Comparison of the wood volumes attacked during the previous year (Initial) and the new attacks between 0 and 100 m from the center of the initial patch.** Each gray line represents one site and the red lines correspond to the average values. There is a strong decrease of the volume attacked even in the control sites without traps.

**Table 1 Analysis of variance table for the Gaussian Linear Model with volume of new attacks as response (log(x+1) transformed).** See also Fig. 5.

|  | Sum Sq | Df | F value | *p*-value |
|---|---|---|---|---|
| Treatment | 11.928 | 2 | 2.738 | 0.069 |
| Year | 1.393 | 1 | 0.640 | 0.425 |
| Treatment x year | 16.830 | 2 | 3.863 | 0.024 |
| Spruce area within 100 m | 5.268 | 1 | 2.419 | 0.123 |
| Initial volume attacked | 3.907 | 1 | 1.794 | 0.183 |
| Residuals | 252.655 | 116 | – | – |

Note:
Spruce area and initial volume are square root transformed and centered on their mean.

of the control sites in 2020 and 40% in 2021 had no new attacks). The global trends seem to be similar between both years despite the fact that the epidemic phase at the regional scale reached a peak in 2020 with a decrease of the levels of attack in 2021 (*OWSF, 2021*).

## Effect of trapping on wood volumes attacked

Table 1 summarizes the statistical analysis for the volume of new attacks (see Supplements, section 3.1 for details). We found no significant association between the volume of new attacks and the initial volume attacked nor the available area of spruce within 100 m. The year x treatment interaction is however significant ($F_{2,116} = 16.8$, $p = 0.02$) which means that there are differences between the treatments but they are not the same for both years. So, we computed all pairwise comparisons between treatments within each year. These results are summarized on Fig. 5 with letters (see also Supplements, section 3.1.3 for details).

In 2020, the volumes of new attacks in crosstraps sites (95% Confidence Interval CI [3.4–14.2 m³]) were slightly but significantly ($t = 2.71$, $p = 0.04$) more important than in

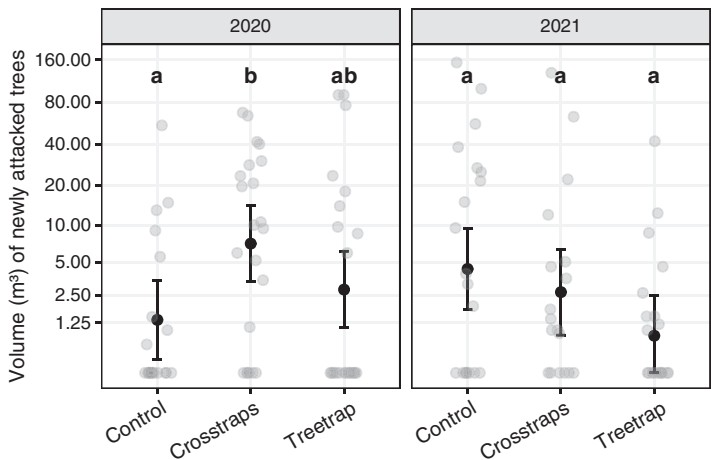

**Figure 5 Comparison of the wood volumes of new attacks between treatments for each year.** The dark circles and error bars represent the linear model predictions and their 95% Confidence Intervals. The gray circles represent the observed values. The letters summarize the *post-hoc* all pairwise comparisons: different letters indicate a significant difference (after *p*-value correction for multiple testing). These comparisons are valid only within a year. The volumes of new attacks were higher in crosstraps sites relative to the control sites but only in 2020.

**Table 2 Analysis of Deviance table (Likelihood Ratio – LR – tests) for the binomial GLM with presence/absence of new attacks as response.** See also Fig. 6.

|  | LR | Df | *p*-value |
|---|---|---|---|
| Treatment | 8.568 | 2 | 0.014 |
| Year | 0.094 | 1 | 0.759 |
| Treatment x year | 1.321 | 2 | 0.517 |
| Spruce area within 100 m | 7.952 | 1 | 0.005 |
| Initial volume attacked | 1.576 | 1 | 0.209 |

**Note:**
Spruce area and initial volume are square root transformed and centered on their mean.

control sites (95% CI [0.2–3.5 m³]). We observed no significant differences with treetrap sites (95% CI [1.1–6.2 m³]) (Fig. 5). In 2021, we found no significant differences between the volumes of new attacks in tree-trap sites (95% CI [0–2.5 m³]), crosstraps sites (95% CI [0.9–6.4 m³]) and control sites (95% CI [1.8–9.4 m³]) (Fig. 5).

## Effect of trapping on presence/absence of new attacks

Table 2 summarizes the statistical analysis for the proportion of sites with new attacks (see Supplements, section 3.2 for details). We found a positive association with the area of available spruce within 100 m (Likelihood Ratio LR = 7.95, df = 1, *p* = 0.005) but no association with the initial volume attacked. The year x treatment interaction and year main effect are not significant but we found a significant treatment main effect (LR = 8.57, df = 2, *p* = 0.014) which means that there are differences between treatments but that these differences are similar between years. So, we computed all pairwise comparisons between all treatments for both years together (using the averaged predictions for both years).

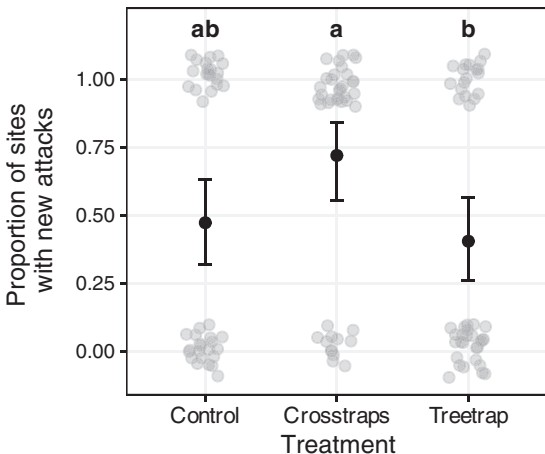

**Figure 6** **Comparison of the proportion of sites with new attacks between treatments.** The dark circles and error bars represent the binomial GLM predictions and their 95% Confidence Intervals. The gray circles represent the observed values (here: presence/absence of new attacks + random noise to limit over-plotting). The results were pooled for both years because of the absence of year x treatment interaction. The letters summarize the *post-hoc* all pairwise comparisons: different letters indicate a significant difference (after *p*-value correction for multiple testing). The proportion of sites with new attacks is higher for crosstrap sites compared to tree-trap sites.

These results are summarized on Fig. 6 with letters (see also Supplements, section 3.2.3 for details).

New attacks are significantly (z = −2.64, p = 0.02) more frequent in crosstraps sites (95% CI [56–84%] of sites with new attacks) than in tree-trap sites (95% CI [26–57%]). The control sites show intermediate predicted values (95% CI [32–63%]) with no significant differences with the two other treatments (Fig. 6).

## DISCUSSION

We found no evidence that the presence of trapping devices in forest stands previously infested by *Ips typographus* could contribute to reducing the probability nor the intensity of new attacks. On the contrary, sites with crosstraps seemed to show increased risk of new attacks. On the other hand, we globally observed a substantial damage reduction in all three treatments compared to the initial infestation levels and even no new attacks at all for half of the control sites. We also found no correlation between the initial levels of attack and the volume or probability of new attacks.

Several non-mutually exclusive hypotheses may explain why the traps did not contribute to damage reduction in our experimental sites. One might wonder whether our traps were fully functional. The average catches of our crosstraps (2,267 and 743 beetles per trap per week in 2020 and 2021, respectively) were similar or greater than average catches reported in the literature with standard pheromone traps active during epidemic periods (between 600 and 1,700 beetles per trap per week; *Raty et al., 1995*; *Faccoli & Stergulc, 2008*; *Galko et al., 2016*; *Grodzki, 2021*). This indicates that both the crosstraps and the attractants were functional, attracting and catching large numbers of beetles. We did not record the number of beetles caught by the tree-traps since this is less straightforward than

for the crosstraps and it was not the goal of this study. However, given that tree-traps were baited with the same quantity of attractants as the crosstraps, and that almost no galleries were found under their bark (indicating that the insecticide was active during the entire trapping period), we can assume that the tree-traps were also fully functional. Even though our traps collected similar quantities of beetles than other studies, it might still be negligible relative to the large number of beetles present in the environment during this severe outbreak. These beetles may originate from the clear-cut (overwintering beetles), but also from other stands further away (dispersing beetles flying over the clear-cut by chance). *Ips typographus* has good dispersal capacities (*Valeria et al., 2016*; *Ellerstrand et al., 2022*; *Müller et al., 2022*) and many specimens from foreign forest stands are likely to be attracted by pheromone traps when transiting by the experimental patches (*Weslien & Lindelöw, 1989*). It is also possible that sanitation felling is so effective to reduce the new attacks that adding traps makes little difference.

Not only the presence of trapping devices did not reduce *I. typographus* infestations, but sites with crosstraps also showed a higher probability of being attacked (56–84% of crosstrap sites with new attacks according to our model predictions). As stated above, it seems that the pheromones attracted additional beetles to the patches (or prevented local beetles to disperse) but that the crosstraps were not efficient enough to neutralize them, inducing attacks on the surrounding trees. This could be due to the fact that, once in the vicinity of the traps, a combination of visual cues and random processes may also cause at least some of the attracted bark beetles to land and initiate attacks on real trees (*Campbell & Borden, 2006*; *Saint-Germain, Buddle & Drapeau, 2007*). An alternative explanation could be the saturation of the traps. The crosstraps collectors were emptied every week and were almost never totally filled with beetles. However, during flight peaks with epidemic population levels, thousands of beetles are concomitantly attracted by the traps, potentially briefly saturating the collectors: if too many specimens fall at the same time in the collector, a supernatant layer is likely to form, preventing newly trapped beetles from drowning and allowing them to fly away. In addition, the volatiles released by decomposing beetles may mask pheromones and further reduce trap efficiency (*Zhang et al., 2003*). In comparison, tree-traps are probably more attractive—because closer to natural trees—and less prone to saturation and even if some beetles fly away, they may have been exposed to enough insecticide to die before damaging standing trees. Previous studies indeed reported a higher efficiency of tree-traps (*i.e.*, whole tree sprayed with insecticide and baited with pheromones) compared to conventional pheromone traps (between 1.7 and 13 times more effective depending on the study; *Abgrall & Schvester, 1987*; *Drumont et al., 1992*; *Raty et al., 1995*; *Grégoire et al., 1997*). These efficiency estimates were however based on the number of insects caught, not on the damage caused to surrounding trees.

In most experimental sites, including the controls, we observed a medium to strong reduction in attacked volumes (Fig. 4) relative to the previous year. This could be due to a natural decrease of the epidemic pressure at a regional level independent from our experimental treatments. Indeed, the available epidemiological data shows that the total volume of spruce attacked by *I. typographus* all over Belgium peaked in 2020 and started to decrease in 2021, probably thanks to an unusually rainy weather favorable to trees but

 

unfavorable to bark beetles (*OWSF, 2021*). However, our study took place during two different years and we observed the same strong decrease of attacks in both years, including during the 2020 year where we should have observed similar or even higher levels of attacks relative to 2019. In the control sites, we even had higher rates of sites without any new attacks in 2020 (60%) than in 2021 (38%). The global epidemic trend thus appears a poor predictor of the observed local pattern. In line with this, it was reported that a large proportion of small infestation spots may go extinct from year to year with very little or no sanitation cutting effort (*Kärvemo et al., 2016*; S. Kärvemo, 2022, personal communication). An alternative explanation could be that the sanitation felling alone was efficient enough to remove the majority of the local bark beetles and prevent most of the new attacks. We have no way of explicitly testing this hypothesis with our experimental design as we have not included any comparison site without sanitation felling. We didn't include such negative controls in our study to keep the number of treatments low (and increase the statistical power), because of legal reasons (removing attacked trees was a legal obligation in Southern Belgium up to mid 2020) and also because it would have been very difficult to find forest managers who would accept to take the risk of not removing attacked trees. Indeed, sanitation felling and salvage logging are the most widely recommended management strategies against *I. typographus* (*Wermelinger, 2004*; *Fettig & Hilszczański, 2015*; *Hlásny et al., 2019*). Several studies have shown that the presence or the intensity of salvage logging and/or sanitation felling are associated with reduced *I. typographus* damages (*Wichmann & Ravn, 2001*; *Schroeder & Lindelöw, 2002*; *Stadelmann et al., 2013*; *Havašová, Ferenčík & Jakuš, 2017*; *Miścicki & Grodzki, 2021*). *Potterf et al. (2019)* also showed that without any intervention (*i.e.*, no management practices to reduce beetle impacts), infested patches tend to grow over time, mostly during the peak and the decline of an outbreak. This result is in line with several studies (including the current one) showing that most of the new attacks are concentrated within a radius of 100 m around previously infested patches, both in intervention (*Wichmann & Ravn, 2001*, the present study) and non-intervention zones (*Kautz et al., 2011*; *Potterf et al., 2019*). Other authors, however, reported a different trend, concluding to a negligible impact of sanitation practices on bark beetles damages, or even to a higher risk of attack in the resulting clear-cuts due to an increased vulnerability of the trees standing near stand edges (*Grodzki et al., 2006*; *Mezei, 2017*; *Vanická et al., 2020*; regarding the higher risk at stand edges, see also *Schroeder & Lindelöw, 2002*).

It is difficult to draw definitive conclusions from these contradictory studies, which use an observational rather than experimental approach (more experimental studies with randomized treatment allocation would be difficult or impossible to implement in most cases, for practical or legal reasons). Most studies compare for example large intervention and non-intervention areas from different regions that may differ in many other factors than sanitary practices (*e.g.*, non-intervention nature reserves compared with stands managed for timber production). Some studies also lack true replicates and most investigate the sanitary impact at the landscape scale, while our study focuses on a more local effect at the stand scale.

Interestingly, we found no correlation between the wood volumes initially attacked in the patches and the new attacks recorded during the next year. If sanitary felling does

participate in damage reduction and is correctly implemented, most of the local beetles from the preceding season should be neutralized. Therefore, this absence of correlation is again compatible with the hypothesis of a strong effect of sanitation felling. This absence of correlation may also be partly due to the fact that we considered the initial volumes attacked over the whole preceding year (as detailed data over time was not available) rather than the last attacks performed before the winter. Only the last generation of beetles from the previous year is susceptible to participate in new attacks the subsequent year.

Our results differ from those of *Faccoli & Stergulc (2008)* who conducted a similar experimental study on the effect of trapping on damage reduction at the stand scale in the Italian Alps. During a whole active season of *I. typographus*, they monitored new attacks on standing trees around previously infested patches. Three types of trapping devices, Theysohn slot-traps, standing trap-logs and lying trap-logs, were compared to control sites without traps. They observed a remarkable decrease of more than 80% of the attacked volumes in all sites with trapping devices relative to the previous year. By contrast, the new infestation levels remained similar to the initial ones in the control sites. They concluded that traps (all three types tested) efficiently reduce damages caused by *I. typographus* in previously attacked stands. In our study, we also observed a strong reduction of attacked volumes, but this reduction concerned sites with traps as well as control sites (See Supplements, section 4 for a comparison of both datasets). The question then arises as to why similar experimental set-ups have produced such different results. Several, not mutually exclusive, explanations may account for this discrepancy: differences in sanitation practices, in the trapping period, in epidemic levels, in the type and density of the trap used, in climatic and environmental conditions.

First, and perhaps most importantly, the sanitation practices differed between the two studies. In both studies, the trees initially attacked during the previous year have been cut and removed before the beginning of the monitoring. In our study, sanitation felling and removal of newly infested trees were then performed throughout the monitoring year, whereas in the study of *Faccoli & Stergulc (2008)*, newly attacked trees were left in place (M. Faccoli, 2021, personal communication). In the latter case, the second generation (two generations per year in the Italian Alps at the altitude of the study; M. Faccoli, 2021, personal communication) of local bark beetle populations emerging from infested trees should therefore be higher than with regular sanitation felling (assuming that sanitation is performed on time). These beetles may have initiated new attacks in the control sites but could have been partially neutralized in sites with traps (see below). The lack of damage reduction observed in the six control sites by *Faccoli & Stergulc (2008)* strengthens the hypothesis that, in our study, sanitation felling is responsible for the decrease in attacked volumes. The damage reduction induced by the continuous sanitation felling might already be so important that adding traps has little effect in our study. However, in the present study, we observed no new attacks in 60% (2020) to 42% (2021) of the control sites. In these sites, there were thus no differences in sanitation practices between the two studies since no further sanitation was required after the initial clearing. A difference in sanitation practices is therefore insufficient to explain the differences in attacked volumes between the two studies.

Second, the traps were not active during the same period between the two studies. In our study, the traps were active from the beginning of April to the beginning of July and aimed at neutralizing the first generation of beetles. We wanted to capture as much as possible the overwintering bark beetles from the patch that could have escaped sanitation felling, while limiting the risk of attracting subsequent generations of dispersing beetles from outside the patch. In the study of *Faccoli & Stergulc (2008)*, the traps were active during the whole activity period of *I. typographus* (from the end of April until mid-September in the Italian Alps). This prolonged period of trapping may have reduced the attacks of the second generation of beetles, especially in the absence of sanitation felling. The risk of attracting beetles from outside the patch or of saturating the traps with too many beetles could have been lower in the Italian context given the different epidemic level (see below). In the present study, the higher levels of attacks observed in sites with crosstraps placed only in the spring suggest that it would have been indeed risky to extend the trapping period over the whole year at a time when the amount of dispersing beetles from outside the patch is highest. However, this is only a working hypothesis and the efficiency of spring *vs.* whole year trapping could be explicitly compared in future studies.

Third, the epidemic level experienced during the study in Belgian spruce forests was much higher than that reported in the Friuli Venezia Giulia region. *Faccoli & Stergulc (2008)* mentioned attacked wood volumes of 0.12 m³/ha in 2004, whereas it exceeded 7 m³/ha in 2019 in the Walloon region (*Alderweireld et al., 2015*; *OWSF, 2021*). In the Belgian context, the traps could have been less able to neutralize the overwhelming quantity of beetles present in the environment.

Fourth, the trapping treatments were not identical between the two studies. The efficiency of the pheromone traps used (Theysohn slot traps *vs.* crosstraps) is assumed to be similar (*Galko et al., 2016*) but our tree-traps and the Italian trap-logs differed. We used whole timbers baited and sprayed with insecticide as tree-traps. *Faccoli & Stergulc (2008)* tested standing and lying trap-logs of 1.5 m long, also baited and sprayed with an insecticide. However, whole timber tree-traps are expected to be more efficient than trap-logs (*Faccoli & Stergulc, 2008*). In addition, they tested multiple trap densities relative to the initial wood volume attacked: 15, 30 or 40 m³/trap (15 m³/ trap represents the highest trapping intensity). In our study, we invariably placed three crosstraps or one tree-trap per site. The density of trapping was nevertheless changing between sites because the initial volume was not constant. Overall, the trapping density in our study was higher or comparable to the Italian study for the crosstraps (range 3–30 m³/trap; see Supplements section 4.1). Although the tree-traps and trap-logs are probably not directly comparable, the densities of our tree-traps covered the whole density range tested by Faccoli and Stergulc (range 5->100 m³/trap). Differences of trap types or density are thus unlikely to explain the differences in results between both studies.

Fifth, the study of *Faccoli & Stergulc (2008)* was performed in the Italian Alps, while our sites were distributed in the Belgian Ardennes, two regions with rather different climatic and landscape contexts. In the southern part of Norway spruce distribution range, elevation is an important factor for its growth due to colder temperature and water balance (*e.g.*, *Seynave et al., 2005*). Therefore, mountainous conditions of the Italian Alps should

better fit the requirements of spruce trees and enhance their resistance to bark beetle attacks. Under these conditions, even a limited effect of trapping may significantly improve damage reduction if fewer insects need to be caught to reduce spruce mortality.

Finally, we cannot rule out the possibility that the reduced number (six) of control sites from the Italian study displayed no damage reduction by bad luck alone, even with low $p$-values (false positive, type I error). Among our 40 control sites, several indeed showed similar or higher damage levels compared to the initial wood volume attacked – as in the Italian study – and could have been selected with a lower number of replicates (Fig. 4). This kind of study is quite difficult to carry out: multiple factors may influence the levels of new attacks and induce a high variability between sites and it is also costly to repeat the observations on a large number of independent replicates. High variability and low number of replicates lead to low statistical power which can in turn induce (1) an important risk of publication bias due to a higher frequency of negative results which are less likely to be published, (2) true positive results limited to very large size effects and (3) a higher frequency of false positive results relative to true positive results in the published literature (*Møller & Jennions, 2001*; *Mogil & Macleod, 2017*). This could explain the surprisingly small number of similar studies published in the scientific literature on a subject of such economic importance. Another possible reason for the scarcity of the scientific literature is that many of such studies—conducted in collaboration with local administrations—might be published in the gray literature in various national languages difficult to understand for the broader scientific community. For example, a recent Swedish study (*Larsson et al., 2021*) seems to have a very similar approach to the present study and to the study of *Faccoli & Stergulc (2008)*. They compared the number of attacked trees between control sites and sites equipped with two types of traps (30 sites in 10 blocks repeated for 2 years). They found high variability in attack levels between blocks and lower but not significant attack levels in sites equipped with traps. However, these results seem so far to have been only published in a working report in Swedish with a short English summary. It is therefore difficult to clearly understand the methodology used and to make an informed comparison with the present results.

## CONCLUSION

Overall, this study provided new insights into the efficacy of different management methods to mitigate the damages caused by *Ips typographus* outbreaks. Our data does not support the use of pheromone traps or tree-traps during spring as a complementary control method in previously infested patches to improve the protection of trees in the vicinity of the initial infestation. The use of baited crosstraps even appears hazardous and could increase the occurrence of new attacks by attracting beetles but failing to neutralize them. The exact impact of sanitation felling on damage reduction remains to be confirmed, but it likely contributed to the important decrease in attacks globally observed in our experimental sites. Interestingly, our results differed from a similar study on the impact of traps in reducing local damages. This highlights the importance of the context in which each experiment is performed, and the necessity of repeating studies, with an appropriate number of replicates and proper controls, before generalizing their conclusions.

## ACKNOWLEDGEMENTS

We thank many persons for their valuable help during this project: Numerous forest officers from the Walloon Forest administration (DNF), L. Sagehomme and the staff from the CRA-W entomology lab for field data collection and their help in the set-up of experimental devices; M. Deproft (CRA-W), D. Jacques and D. Arnould (DNF) for launching this project and for stimulating discussions about the results; J.C. Grégoire (ULB) and Q. Leroy (DEMNA-OWSF) for useful input on the project and C. Lucau-Danila (CRA-W) for GPS data cleaning. We also would like to thank the reviewers for taking the necessary time and effort to review the manuscript.

### Funding

This work was supported by the Walloon Region (No. 2016015 and 2112607). The funders had no role in study design, data collection and analysis, decision to publish, or preparation of the manuscript.

### Grant Disclosures

The following grant information was disclosed by the authors:
Walloon Region: 2016015 and 2112607.

### Competing Interests

The authors declare that they have no competing interests.

### Author Contributions

- Alexandre Kuhn conceived and designed the experiments, performed the experiments, analyzed the data, prepared figures and/or tables, authored or reviewed drafts of the article, and approved the final draft.
- Louis Hautier conceived and designed the experiments, authored or reviewed drafts of the article, provided adequate material/staff to perform the study, and approved the final draft.
- Gilles San Martin conceived and designed the experiments, performed the experiments, analyzed the data, prepared figures and/or tables, authored or reviewed drafts of the article, and approved the final draft.

### Data Availability

All the supplements and data and R scripts needed to reproduce our results are available at figshare: San Martin, Gilles; Kuhn, Alexandre (2022): Do pheromone traps help to reduce new attacks of *Ips typographus* at the local scale after a sanitary cut? Supplements and detailed statistical analysis. figshare. Dataset. https://doi.org/10.6084/m9.figshare.19940321.v1.

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
