# Peer review of "Do pheromone traps help to reduce new attacks of Ips typographus at the local scale after a sanitary cut?"

_PeerJ, doi:10.7717/peerj.14093_

## Round 0.1 · original submission · Major Revisions

Dear Dr. Kuhn and colleagues:

Thanks for submitting your manuscript to PeerJ. I have now received two independent reviews of your work, and as you will see, the reviewers raised some concerns about the research. Despite this, these reviewers are optimistic about your work and the potential impact it will have on research studying spruce bark beetle control. Thus, I encourage you to revise your manuscript, accordingly, taking into account all of the concerns raised by both reviewers.

Reviewer 1 expressed several doubts that could affect the final conclusions. These concerns should be addressed in order to modify the conclusions and/or explain and justify your approach. Reviewer 2 had more minor concerns, though these also need to be addressed, including missing references and methodological details.

I look forward to seeing your revision, and thanks again for submitting your work to PeerJ.

Good luck with your revision,

-joe

Reviewer 1 ·

Basic reporting

The revised MS concerns a very important topic – effective control of the spruce bark beetle Ips typographus infestations in the stands subjected to sanitary felling (the approach usually applied in the stands threatened/attacked by bark beetles). The text is written clearly and correctly, thus easy to understand. The background is satisfactory, used literature properly selected and referenced, although some additional references seem to be necessary in order to improve the background (see additional comments). The structure of revised MS is appropriate for original articles (general), as well as for PeerJ standards. The figures are appropriate, but the quality seems to be too low (maybe in the version accessible for reviewer). The raw data supplied by Authors.

Experimental design

The content of the MS is quite narrow, dealing with specific topic related to forest management, which could be considered as fitting in broadly understood environmental sciences (if one would relate it with the categories listed in the Scope of the journal). Therefore it seems that it could be appropriate rather for the journal dealing with forestry and/or forest management practices. The research question is clearly defined, relevant (see Basic reporting) and meaningful. The Authors correctly point out that the question of the efficiency of pheromone traps is relatively poorly represented in the literature, therefore the expected findings could fil the knowledge gaps concerning this topic. The experimental design is generally correct, however the doubts concerning the use of pheromone traps arise (see Validity of findings). The field investigations were performed rigorously and the analysis of results is appropriate. The description of methods is sufficient for the replication of performed experiments.

Validity of the findings

The findings are based on a large set of data, although containing a considerable share of negative results. The main doubt concerns the discrepancy between the period of pheromone traps exposition (spring) and the period in which the effect (infestation of trees by bark beetles) of experiment was assessed (the entire growing season from spring till autumn). The attacks of the spruce bark beetle occur during the entire growing season, but the pheromone traps were exposed only in spring. It is quite possible that the new attacks occurred in the period after the exposure of traps, thus the effect of pheromone application could be then limited or even nullified. This could be an important cause of obtained results, as the Authors did not provide any data on the timing of new infestations in experimental stands (“the evolution of attacks throughout the year was not recorded”). It is important as the Authors declare the occurrence of several I. typographus generations within a year. The attempt to explain this discrepancy (traps vs. infestations) is not satisfactory. The comparison with the results of Faccoli & Stergulc (2008) seems to be not justified, as those Authors used the traps that were exposed during the entire growing season. On the other hand the Authors correctly point up that the pheromone baits could have the attraction effect resulting in the infestations of standing trees (in the way of beetles’ flight). Finally, the conclusions concerning the use of pheromone traps (no difference between stands with and without pheromones) seem to be not enough justified – the true conclusion, although not directly based on the presented results, is this concerning the efficiency of sanitary felling in the control of bark beetle infestations.

Additional comments

Line 360-361: the Authors state that on the tree-traps “almost no galleries were found under their bark” – it is obvious, as the tree-traps were sprayed with insecticide, which disable the installation of galleries;
lines 365-368: it is true, but some background is necessary – try to provide the references concerning the effective distance of attraction (by pheromones). Such references should also be provided in order to justify the distances used for the assessment of new infestations (lines 222-224).

Reviewer 2 ·

Basic reporting

The language is of good quality, introduction and background describes the research area and importance in a good way and figures and tables are informative and easy to understand. In addition, detailed information of data and analyses are given in Supplements, which is valuable.

Experimental design

The research question is well defined and highly relevant because of the large economic impact of the spruce bark beetle during outbreaks. Only one similar study on the spruce bark beetle have earlier been published. The experimental set up is of high quality with large number of replicates. The methods are described in detail and easy to follow.

Validity of the findings

Regarding the fact that only one earlier similar study have been published, this manuscript makes a valuable contribution. Especially because the earlier study was poorly replicated and gave a somewhat controversial result. As stated in the manuscript there is always a large interest in using pheromone-baited traps as a control method against the spruce bark beetle during outbreaks (both in protected areas and production forests). Thus, it is very important that studies are conducted, and published, that evaluate the efficiency of traps to reduce damages (because traps are not without costs). The detailed discussion about the difference in result compared with the earlier study is very valuable in this context. The statistical analyses seem to be sound. Conclusions are well stated.

Additional comments

I do not have any major concerns regarding the study. The most important comments are:
(1)If possible, it would be nice to give data on when the sanitation cuttings were conducted the year before the experiment was conducted. This because the timing influences the efficiency of the cuttings and thus also the local population at the sites when the experiment was started (see comment below). The same information would be valuable to give for the sanitation cuttings conducted during the summer of the experiment.

(2)If you have data on the distances between the pheromone-baited trapping devices (traps or trap logs) and the closes stand edge with spruce it would be a valuable contribution to the manuscript. Especially because you mention the risk of pheromone baits inducing attacks on the stand edges.

Detailed comments:
Line (L) 43-44: The last sentence could be removed from the abstract. You do not have hard data on this.
L 79-80: I am not sure what you mean by “voluntarily stressed”?
L 83: “Other shortcomings….” Strange sentence
L 154-160: You state that population levels are lowest in spring and highest in summer when second generation beetles have started to fly. I think it is hard to know when population levels are lowest (or highest). If there were hibernating populations in the sites despite sanitation cuttings the populations at the local scale may actually have been highest in spring. I think the argument that tree vitality may be highest in spring is enough as an argument.
L 157 “…capturing as much as possible of all remaining…”
L 156-158: Here I suggest that apart from reducing the local populations that have hibernated at the site with the traps you also may reduce/capture some “immigrating” beetles (originating from other localities).
L 169-171: If you have information about when this was done, I suggest that you add this information. This because the effect of sanitation cutting will vary with the season being most efficient during summer when offspring still in larval or pupal stage (because they die in bark that fall of during tree harvest while adults survive) and before the new generation have started to emerge. Thus, even if you do not have any detailed information, the information you have will still be valuable.
L 196-197: I am not sure what this means? Did you randomly assign treatments to locations within forest district subunits? Or did you also within districts include a geographic element when doing the randomization?
L 206-208: If you could give information about when these cuttings were conducted, and in how many of your sites, it would be valuable information.
L 229-232: Maybe reformulate: “…, the wood volume of new attacks was recorded for the entire period….”
L 240-249: It is nice that you present all these data in detail (in Supplements). One variable that I miss is distance (in meter) from the closes trap to stand edge of spruce forest. Would have been interesting also to include in the analyses because of the risk of pheromone baits inducing attacks on the tress in the stand edge facing the traps (as you also mention in the manuscript). If you do not have detailed information about this, maybe you could give some kind of rough intervals (without analyses).
L 269: Something missing in the parenthesis? Otherwise, remove “…”
L 317: Regarding available area of spruce within 100 m. Did it happen that all spruce, or almost all, within 100 m was killed (i.e. resource depletion) for some localities? If so, that should be mentioned because it could have reduced potential tree mortality. If not, that should also be mentioned.
L 363-364: I am not sure what you mean with “relative to…..present in the environment”? But, few beetles caught in relation to the number of beetles colonising trees within 100 m in most places where attacks occurred. Hard to know how many beetles that moved through the area within a 100 m radii.
L 364-365: Very unlikely that unattacked trees would be more, or similar, attractive as the traps! There is no evidence of a strong primary attraction (i.e. attraction to host volatiles alone) for I. typographus. See e.g. Lindelöw et al. (1992) Can.J. For. Res. 22: 224-228; Schroeder (2003) Forest Ecology and Management 177: 301-311.
L 365-366: This possibility is hard to evaluate without knowing how and when the sanitation felling was conducted. If sanitation felling was conducted with harvesters, a considerable amount of the bark will fall off and remain in the forest with the bark beetle offspring. See two publications about this:
Weslien et al 2022 (in Swedish with English summary): https://www.skogforsk.se/cd_20220316105614/contentassets/16ac29a0cf0f4ad6a5508e1640bdaae1/arbetsrapport_1110-2022_.pdf
Delb et al. (2021) Infektionsgefahr durch Buchdrucker (Ips typographus) aus mechanisch mit Vollerntern aufgearbeiteten Fichten – ein Beitrag zur Entscheidungsfindung in der Praxis. Forschungsbericht FVA-Waldschutz, 31 p.
If the offspring still is in the larval/pupal stage they will not be able to fulfil development. But, if in adult stage they may survive in the bark until next season, at least if trees are harvested in autumn/winter (see ).
L 374-376: See comments above about these factors.
L 378-381: But still, the catches seemed “normal” compared with earlier studies as you stated above.
L 400-401: I am not surprised. It is not unusual that untreated infestation spots go “extinct” between years and that new spots are formed. Extinction seems to be especially probable for smaller spots (see Kärvemo et al. 2016. Local colonization-extinction dynamics of a tree-killing bark beetle during a large-scale outbreak. ECOSPHERE Volume 7 (3)).
L 411-414: See comment above.
L 422-423: Regarding risk of attacks at stand edges facing clear-cuts see also Schroeder & Lindelöw 2002. Attacks on living spruce trees…..Agricultural and Forest Entomology 4: 47-56.
L 453-455: I suggest that you here also mention that there was few replicates in their study because you discuss it later on and it is an important factor to consider.
L 502-506: I suggest that you include this as a separate fifth paragraph/explanation. It is a valid concern.
L 510-515: I agree with all these suggestions. An additional one is that these kind of more “practical/management” studies unfortunately often only are reported in national reports (i.e. grey literature), in best case with a summary in English. One such example of a good trapping study, similar to the one presented here and with the same result, conducted in Sweden can be found in: https://www.skogforsk.se/cd_20210419092300/contentassets/252b792162df4bbe860a46c1f51275ba/arbetsrapport-1078-2021.pdf

---

## Round 0.2 · Minor Revisions

Dear Dr. Kuhn and colleagues:

Thanks for revising your manuscript. The reviewers are very satisfied with your revision (as am I). Great! However, there are a few issues to entertain. Please address these ASAP so we may move towards acceptance of your work.

Best,

-joe

Reviewer 1 ·

Basic reporting

First, I have to repeat from my previous review:
The revised MS concerns a very important topic – effective control of the spruce bark beetle Ips typographus infestations in the stands subjected to sanitary felling (the approach usually applied in the stands threatened/attacked by bark beetles). The text is written clearly and correctly, thus easy to understand. The background is satisfactory, used literature properly selected and referenced, although some additional references seem to be necessary in order to improve the background (see additional comments). The structure of revised MS is appropriate for original articles (general), as well as for PeerJ standards.
As for the quality of figures - the explanation given by Authors is satisfactory to me.

Experimental design

As stated earlier, the content of the MS is quite narrow, dealing with specific topic related to forest management, which could be considered as fitting in broadly understood environmental sciences. Nevertheless, the decision about the publication in PeerJ is up to the Editors.The research question is clearly defined, relevant (see Basic reporting) and meaningful, the experimental design is generally correct. The doubts concerning the use of pheromone traps, even if still existing for me, are sufficiently explained by the Authors. The description of methods is sufficient for the replication of performed experiments.

Validity of the findings

The findings are based on a large set of data, although containing a considerable share of negative results. The main doubt, concerning the discrepancy between the period of pheromone traps exposition (spring) and the period in which the effect of experiment was assessed, is explained by the Author by adding some updates (new parts) to the MS, as well as to the abstract. I have no other remarks to the text.

Additional comments

The suggestions expressed in this section have been addressed by the Author in the updated MS.

Reviewer 2 ·

Basic reporting

The authors have responded to all my comments/questions in my earlier review of the manuscript so I have nothing to add there. The only thing I found was a few typos. The reference Lindelow should be Lindelöw (line 423, 434 and 742). I. typographus should be in italic (line 491 and several places in references).

Experimental design

No comment

Validity of the findings

No comment

---

## Round 0.3 · accepted · Accept

Dear Dr. Kuhn and colleagues:

Thanks for revising your manuscript based on the concerns raised by the reviewers. I now believe that your manuscript is suitable for publication. Congratulations! I look forward to seeing this work in print, and I anticipate it being an important resource for groups studying spruce bark beetle control. Thanks again for choosing PeerJ to publish such important work.

Best,

-joe